# A single-cell atlas depicting the cellular and molecular features in human anterior cruciate ligamental degeneration: A single cell combined spatial transcriptomics study

**Runze Yang, Tianhao Xu, Lei Zhang, Minghao Ge, Liwei Yan, Jian Li, Weili Fu\***

Sports Medicine Center, Department of Orthopedic Surgery/ Orthopedic Research Institute, West China Hospital, Sichuan University, Chengdu, China

## Abstract

**Background:** To systematically identify cell types in the human ligament, investigate how ligamental cell identities, functions, and interactions participated in the process of ligamental degeneration, and explore the changes of ligamental microenvironment homeostasis in the disease progression.

**Methods:** Using single-cell RNA sequencing and spatial RNA sequencing of approximately 49,356 cells, we created a comprehensive cell atlas of healthy and degenerated human anterior cruciate ligaments. We explored the variations of the cell subtypes' spatial distributions and the different processes involved in the disease progression, linked them with the ligamental degeneration process using computational analysis, and verified findings with immunohistochemical and immunofluorescent staining.

**Results:** We identified new fibroblast subgroups that contributed to the disease, mapped out their spatial distribution in the tissue and revealed two dynamic trajectories in the process of the degenerative process. We compared the cellular interactions between different tissue states and identified important signaling pathways that may contribute to the disease.

**Conclusions:** This cell atlas provides the molecular foundation for investigating how ligamental cell identities, biochemical functions, and interactions contributed to the ligamental degeneration process. The discoveries revealed the pathogenesis of ligamental degeneration at the single-cell and spatial level, which is characterized by extracellular matrix remodeling. Our results provide new insights into the control of ligamental degeneration and potential clues to developing novel diagnostic and therapeutic strategies.

**Funding:** This study was funded by the National Natural Science Foundation of China (81972123, 82172508, 82372490) and 1.3.5 Project for Disciplines of Excellence of West China Hospital Sichuan University (ZYJC21030, ZY2017301).

## Editor's evaluation

This is a valuable study of particular interest to those researchers working in the field. The authors provide a single-cell atlas of healthy and degenerative human anterior cruciate ligament using transcriptomic profiling. The quality of the study was considered valuable as it expands current knowledge.

**\*For correspondence:**
foxwin2008@163.com

**Competing interest:** The authors declare that no competing interests exist.

## Introduction

Skeletal ligaments, an important part of the musculoskeletal system, are defined as the dense connective tissue that spans the joint and then is fixed at both ends of the bone (*Hanhan et al., 2016*). One of the main functions of ligaments is mechanical, as they passively maintain joint stability and assist in guiding those joints through their normal range of motion when a tensile loading is applied (*Frank, 2004*). The knee is the largest and most complicated hinge joint associated with weight bearing in the human body. The anterior cruciate ligament (ACL) is crucial for knee kinematics, especially in rotation, and functions as an anterior/posterior stabilizer (*Fleming, 2003*). ACL degeneration can gradually result in chronic knee pain, instability, and even poor life quality (*Thompson et al., 2015*). It has been reported that aging-related degenerative changes in the ACL might also contribute to OA occurrence and progression (*Loeser, 2010*), but mechanisms of ACL aging and degeneration remain unclear.

The extracellular matrix (ECM) is the physical basis for the biological functions of ligaments, and ACL degeneration is accompanied by alterations in the ECM. The ECM of ACL consists of collagen types I, II, III, and V, elastin, and proteoglycans (*Laurencin and Freeman, 2005*), and collagen type I is a major determinant of tensile strength (*Corps et al., 2006*). ACL degeneration is characterized by the disorganization of collagen fibers, cystic changes, mucoid degeneration, and chondroid metaplasia (*Hasegawa et al., 2012*). Different types of pathological changes correspond to different ECM alterations. For example, mucoid degeneration reflects the degradation of collagen and deposition of new glycosaminoglycans, and cystic changes are devoid of ECM in the diseased area (*Hasegawa et al., 2012*). It has been reported that some regions in the degenerated ACL have decreased type I collagen, whereas type II, III, and X collagen and aggrecan are significantly increased, and this abnormal ECM production can lead to a biomechanical fault (*Hasegawa et al., 2013*; *Hayashi et al., 2003*). In recent years, many scholars have paid attention to the degeneration mechanism of ligaments. TGFβ1, a member of the TGF superfamily, can be used to induce chondrogenic differentiation of ligament-derived stem cells in vitro (*Schwarz et al., 2019*) and is considered to be an essential molecule contributing to chondroid metaplasia. It also has been reported that mechanical stretching force can promote TGF-β1 production by ligament cells, contributing to ligamental hypertrophy (*Nakatani et al., 2002*). Some reports suggested that complement cascade could lead to ECM catabolism and contribute to ligament degeneration (*Busch et al., 2013*; *Schulze-Tanzil, 2019*). Akihiko Hasegawa et al. suggested that there is inflammation in the degenerative ligament, manifesting as leukocyte infiltration and neovascularization between collagen fibers within ACL substance, and once inflammation is initiated, its synergy with mechanical forces could facilitate ACL degeneration (*Fleming et al., 2005*; *Hasegawa et al., 2012*).

There are various types of cells in the ligament (*Kharaz et al., 2018*), and building a detailed ligamental cell landscape is essential to understanding ligamental characteristics and underlying pathogenesis of ACL degeneration. Currently, the reports on the ACL cellular changes during the process of degeneration mostly focus on chondroid metaplasia and progenitor cells or myofibroblasts recruitment and proliferation (*Hasegawa et al., 2013*; *Mullaji et al., 2008*). However, the cellular changes during ligamental degeneration are still unclear. Here, we performed single-cell RNA sequencing (scRNA-seq) and spatial RNA sequencing (spRNA-seq) to obtain an unbiased atlas of ACL cell clusters. Our findings provide a better understanding of the inherent heterogeneity, construct the classifications of fibroblasts, and profile the spatial information of identified cell clusters in the ACL. Notably, we also identified the cell interactions during the disease process and demonstrated the role of FGF and TGF-β signaling pathways in ligament degeneration. Thus, our study reveals the cellular landscape of the human ACL and provides insight that could help to identify therapeutic targets for human ligamental degeneration.

## Methods

### Human ligament cell sample preparation

The ligament specimens were collected from four joint replacement patients with osteoarthritis (degenerated ACLs) and four amputation patients with severe trauma or malignant bone tumor (healthy ACLs; *Supplementary file 1*).

All specimens removed were placed immediately in Dulbecco modified Eagle medium (DMEM) free of antibiotics and FBS under 4°C. Ligament samples were rinsed in precooled PBS and then

were cut into 1 mm³ pieces. Containing Collagenase type I (2 mg/ml) and 0.25% (w/v) trypsin DMEM was used to digest the specimens for 1 hr at 37 °C using a Thermomixer at 1200 rpm (Eppendorf, Hamburg, Germany). After that, Ham's F-12 media containing 10% FBS was added to stop the process of digestion, and then the digested tissue passed through a 100 μm cell strainer. Finally, after centrifugation, cells were collected for subsequent batch analysis.

## Single-cell RNA-seq analysis

The raw single cell sequencing data was mapped and quantified with the 10×Genomics Inc software package CellRanger (v5.0.1) and the GRCh38 reference genome. Using the table of unique molecular identifiers produced by Cell Ranger, we identified droplets that contained cells using the call of functional droplets generated by Cell Ranger. After cell-containing droplets were identified, gene expression matrices were first filtered to remove cells with >5% mitochondrial genes, <250 or >8000 genes, and <500 UMI. Downstream analysis of Cellranger matrices was carried out using R (4.1.3) and the Seurat package (v 4.1.0, https://satijalab.org/seurat/). In total, 49,356 cells with an average of 2435 genes per cell were selected for ongoing analysis. Of these single cells, 24,721 were obtained from lesions ligament, including L2, L5, L6, and L8. The remainder were cells from healthy and included L1, L3, L4, and L10.

## Single-cell trajectory analysis

We used the Monocle3 v. 2.8.0 R package to infer a hierarchical organization part of fibroblasts to organize these cells in pseudotime. We took these subpopulations from the Seurat data set from which we reperformed shared nearest neighbor clustering and differential expression analysis as described previously. We then selected the differentially expressed genes based on the fold-change expression for Monocle to order the cells using the DDRTree method and reverse graph embedding. We find gene co-expression modules according to the trend of gene expression and show the relationship between cells and modules in the form of heat maps.

## Ligand-receptor interaction model

We used the ligand-receptor interaction database from the cellchat database to determine potential ligand-receptor interactions between fibroblast subpopulations and other cell types. The expression data were preprocessed for subsequent cell-cell communication analysis. The ligand or receptor that is overexpressed in a class of cells is first identified, and the gene expression data is then projected into a protein-protein interaction network. Whenever a ligand or receptor is overexpressed, the ligand-receptor interaction is recognized. CellphoneDB is based on the expression of a receptor in one cell type with a ligand in another cell type, resulting in rich receptor-ligand interactions between two cell populations. For the gene expressed by the cell population, the percentage of cells expressing the gene and the average gene expression were calculated. The gene was removed if it was expressed only in 10% or less of the cells in the population (the default value is 0.1).

## Multiplex immunofluorescence (OPAL) staining

The tissue slices underwent deparaffinization and rehydration, and then heat-induced antigen retrieval was used. Following that, the paraffin slide containing the ligament tissue was subjected to continuous staining using the Opal Polaris Multiple-Color Manual IHC Kit (NEL861001KT), which involved incubating the tissue with different primary antibodies specific to various cell markers. We used different primary antibodies to simultaneously label pericyte (ACTA2), pericyte1 (MYH11), and pericyte2 (CD90) on the same tissue slide. APOE was used to label Fib.9 and TOP2A was used to label Fib.8. DAPI was used to stain cell nuclei. An automated staining system (BOND-RX, Leica Microsystems, Vista, CA) was used to perform the chromogen-based multiplex immunohistochemistry labeling. The Opal Polaris dyes were used to pair with these antibodies containing fluorophores for tyramide signal amplification (TSA) to enhance sensitivity. The following primary antibodies: FGF7 (160128, Abclonal), TGFβ1 (HA721143, HUABIO), APOE (YT0273, Immunoway), TOP2A (YM6914, Immunoway), ACTA2 (ab5694, Abcam), MYH11 (A4064, Abclonal), and CD90 (ab307736, Abcam) were used.

## Spatial-transcriptome analysis

We used L1, and L8 samples for spatial transcriptome analysis. We performed the Seurat standard analysis. Machine prediction and marker gene methods were used to determine the cell type, and SCTransform was used for standard analysis of the data. First, we determine it is specific enough to find a topic profile (similar to a feature vector) for each cell type and then deconvolve the information by cell type and superimpose it onto the slice in pie chart form.

## Deconvolution analysis

SCDC (v 0.0.0.9000) tool combined with single cell data was used to deconvolve bulk RNAseq data for cell type and content proportion analysis bulk RNAseq data. The bulk RNAseq data is also our own data. Histogram and heat map can be drawn for the predicted data, and the proportion of cell type content can be compared between groups. Then, we used GSVA (v 1.42.0) to calculate the same data and draw heat maps to aid the analysis.

## Statistical analysis

A Nonparametric Wilcoxon rank sum test was used to analyze the differences between the two groups. All statistical analyses were performed in R or GraphPad Prism (version 5.0). Statistical significance was defined as $*p<0.05$, $**p<0.01$, $***p<0.001$.

# Results

## Comprehensive scRNA-seq analyses reveal the major cell types in the human normal and degenerated ligament

To decipher cellular heterogeneity and construct the cell landscape for ligament degeneration, we performed single-cell transcriptomic profiling of cells from four healthy and four degenerated ACL samples (*Figure 1—figure supplement 1*). The characteristic pathological findings in the ACL of OA were identified by arthroscopy (*Figure 1A and B*). In the normal group, the ligaments are white stripes with a clear structure, and blood vessels can be seen on their surfaces. In the degenerated group, the ligaments were swollen, the structure was disorganized, and the color of the tissue was dark. From the images of knee MRI, we can also find that the signal of normal ACL on the T2 phase is uniform and the structure is clear, while the signal of the ACL in the degenerated group is mixed, and the clear structure cannot be distinguished (*Figure 1—figure supplement 2A*). After rigorous quality control, we obtained the transcriptomes of 49,356 cells (normal: 24635 degenerated: 24721) and then employed differential gene expression analysis to discern cluster-specific markers. Firstly, we used principal component analysis (PCA) to reduce the dimension, and then we adopted the uniform manifold approximation and projection (UMAP) method to conduct the next analysis. Unbiased clustering based on UMAP identified four cell classes and histrionic lineage-defining genes were detected, including endothelial cells: PECAM1, VWF, and PLVAP (3862, 7.83%), fibroblasts: DCN, LUM, and COL1A2 (32644, 66.14%), immune cells: PTPRC, SRGN, and CD163 (8021, 16.25%), and pericytes: ACTA2, MCAM, and RGS5 (4829, 9.78%) (*Figure 1C–F* and *Figure 1—figure supplement 2B*). According to *Figure 1I*, we could further confirm that the selected cell markers could well identify various cell types. UMAP plots and stacked bar plots in *Figure 1G* described the distributions of cells in the healthy and degenerated samples. We can conclude that, in general, fibroblasts are the main cell type in both normal and degenerated ligaments and the degenerated group had a higher immune cell ratio and lower pericyte ratio than the normal group. We next analyzed the number of differentially expressed genes (Degs) between healthy and degenerated ligament clusters. The results demonstrated that the fibroblast had the largest difference, implying that fibroblasts undergo significant changes during the degenerative process (*Figure 1H*). We also collected normal as well as degenerated ligament specimens for bulk sequencing and performed deconvolution analysis between bulk sequencing results and scRNA-seq results. The results illustrated that scRNA-seq samples were in good agreement with bulk sequencing samples (*Figure 1J and K* and *Figure 1—figure supplement 2C*). After that, through Degs analysis in the scRNA-seq results, we selected genes highly expressed in the degenerative group of the four-cell subsets (fibroblast, immune cell, endothelial cell, and pericyte). By examining the expression of these genes in bulk sequencing data, it can be found that these genes are also highly expressed in the diseased sample of bulk RNA-seq (*Figure 1—figure*

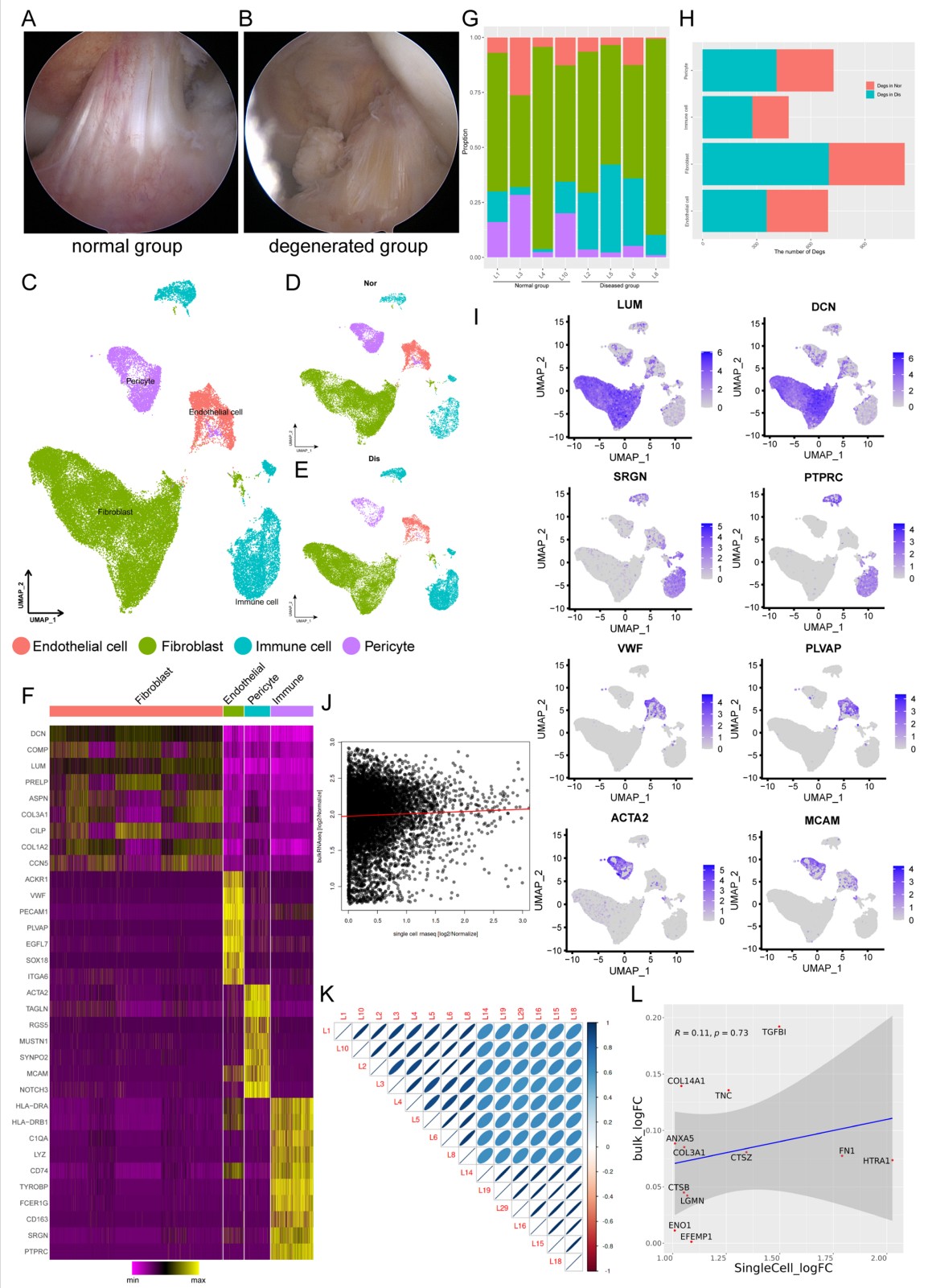

**Figure 1.** Single-cell RNA-seq reveals major cell classes in human ligament. (**A and B**) A photograph of typical normal (left) and degenerated (right) anterior cruciate ligaments (ACL) under arthroscopy. (**C**) UMAP visualization of all cell clusters in collected ligamental specimens. (**D and E**) UMAP visualization of the donor origins in normal/diseased samples. (**F**) Heatmap of selected marker genes in each cell cluster. (**G**) The percentages of the identified cell classes in normal/diseased ligament. (**H**) Number of differentially expressed genes (DEGs) in each cell type of normal/diseased status.

*Figure 1 continued on next page*

*Figure 1 continued*

(**I**) Feature plots of expression distribution for selected cluster-specific genes. Brighter colors indicate higher expression levels. (**J**) Gene expression profiles from bulk samples (n=6) and in single cell samples (n=8) were averaged and plotted on X and Y axes, respectively. Red lines indicate linear model fit and the diagonal. (**K**) Correlation heatmap shows Pearson's correlation between all bulk and in single cell samples. (**L**) Consistency analysis of Degs between normal and diseased groups in single cell and bulk RNA sequencing data.

The online version of this article includes the following figure supplement(s) for figure 1:

**Figure supplement 1.** The overall workflow of our research programme.

**Figure supplement 2.** Typical MRI images of different types of ligaments and consistency analysis between single cell and bulk RNA sequencing data.

*supplement 2D–G*). The results of the direct comparison of gene expression between the bulk RNA-seq and scRNA-seq data per se also illustrated that scRNA-seq samples present high consistency with bulk RNA-seq samples (*Figure 1L*).

## Characterization of fibroblast subpopulations in healthy and degenerative ACL

Because fibroblasts are known to play a vital role in ECM homeostasis and the degeneration of the ligament and undergo remarkable changes during the process of ligament degeneration, we next conducted further analysis of fibroblasts in normal and degenerated ligaments. We subclustered fibroblasts and identified 10 subsets (Fib.1-Fib.10; *Figure 2A*). The distribution shown in *Figure 2B* indicated that the batch effects are resolved well. As illustrated in *Figure 2C*, we obtained the cell proportions of the fibroblast subpopulations between normal and diseased groups. We can conclude that the ratio of fibroblast subclusters varied greatly between these two groups. The ratio of Fib.1, Fib.2, Fib.8, and Fib.9 in the diseased group were higher than that in the healthy group, and the ratio of Fib.3, Fib.4, and Fib.7 in the healthy group were higher compared to the diseased group. The proportion of remaining fibroblast subclusters did not differ significantly between the two groups.

According to the analysis of Degs among these fibroblast subgroups, we tried to reveal the identity of each subpopulation (*Figure 2D*). Combined with the analysis of the proportion of cell subclusters, we inferred that Fib.1 and Fib.2 are ECM remodeling-associated fibroblasts as they highly expressed the genes related to ECM components and ECM decomposition such as COL1A1, CTHRC1, COL3A1, FBLN1, MMP2, MMP14, and TPPP3. From the perspective of ECM synthesis, Fib.1 and Fib.2 are slightly different. Fib.1 highly expressed the collagen-related genes, but Fib.2 highly expressed the genes related to the synthesis of connexins. Fib.3 expressing metallothionein family genes such as MT1X, MT1E, and MT1M is a population of homeostasis-associated fibroblasts with defensive functions. Fib.4, expressing genes associated with the formation and organization of the ligamental ECM such as ANGPTL7, MYOC, CILP2, and THBS1 is a group of resident fibroblasts that function normally. Fib.5 is a cluster of structural fibroblasts within the ligament as they highly expressed multiple collagen-related genes such as COL3A1, COL6A1, COL5A1, COL1A1, and TGFVBI. Fib.6 is a cluster of chondrocytes as they highly expressed cartilage-related genes such as COL2A1, FMOD, CHAD, ACAN, and CILP2. We cannot define Fib.7 well because they expressed Degs without specific tags. Fib.8 is a group of cycling cells as they highly expressed cell cycle-related genes such as CENPF, TOP2A, and MLI67. Fib.9 is a group of inflammation-related fibroblasts as they highly expressed CXCL14, CXCL12, C3, and C7. Fib.10 may have the function of damage repair as they highly expressed anti-inflammation and anti-ECM decomposing associated genes such as PRG4, DEFB1, TIMP3, and HBEGF. To verify the accuracy of the classification of fibroblast subclusters, the top 50 Degs of each subpopulation were deconvolved on the bulk RNA-seq data by the GSVA method. The results illustrated that fibroblast 3,4 scored higher in the normal samples and fibroblast 1,2,9 scored higher in the diseased samples, which were consistent with the changes in proportion of scRNA-seq.

For further in-depth investigation of changes in ligamental fibroblasts during the degeneration process, we performed the Degs analysis in the whole fibroblast population between the normal and degenerated ligaments. We observed genes upregulated in the degenerated group such as CFD, SPP1, COL1A2, and COL3A1, which were related to scar healing and ECM remolding (*Figure 2E*). Gene Ontology (GO) and Kyoto Encyclopedia of Genes and Genomes (KEGG) analysis were conducted for further functional interpretation of these Degs. The results illustrated that the upregulated genes in the diseased group were enriched for the terms 'extracellular region', 'TGF-beta signaling pathway'

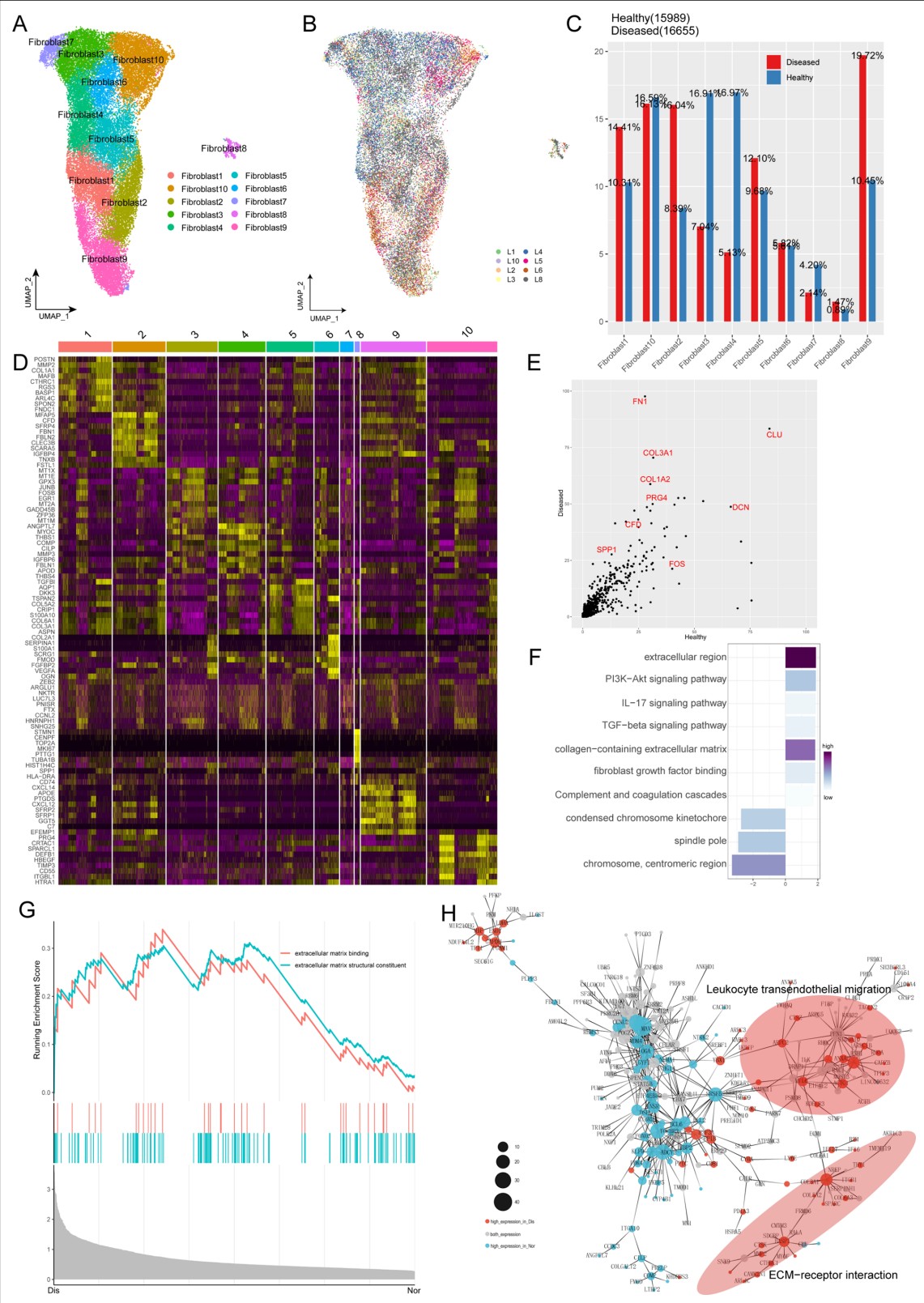

**Figure 2.** Characterization of fibroblast subclusters across different statuses in human ACL. (**A**) UMAP visualization of the subclusters of fibroblasts. (**B**) UMAP visualization of the distribution of fibroblast subclusters at different samples. (**C**) The proportions of 10 fibroblast subpopulations in normal and diseased ligaments. (**D**) A cell-level heatmap reveals the normalized expression of DEGs for each fibroblast cluster defined in. (**E**) Volcano plot showing the DEGs between two statues of ligamental fibroblasts. The x axis represents highly expressed genes in normal cells, and the y axis represents

*Figure 2 continued on next page*

*Figure 2 continued*

highly expressed genes in diseased cells. (**F**) GO and KEGG enrichment analysis of DEGs in ligamental fibroblasts between normal and degenerated states. (**G**) GSEA enrichment plots for representative signaling pathways upregulated in fibroblasts of diseased samples, compared with normal samples. (**H**) Gene-gene interaction networks between DEGs in ligamental fibroblasts of normal group and fibroblasts of diseased group.

and 'Complement and coagulation cascades' (*Figure 2F*). Gene Set Enrichment Analysis (GSEA) also demonstrated that ECM-associated signaling, ECM binding and ECM structural constituent, were activated in the degenerated group, which implied that the changes of ECM play an important role in the ligament degeneration (*Figure 2G*). And then, by building gene interaction networks, we identified that genes upregulated in degenerated groups were concentrated in two parts, ECM-receptor interaction and leukocyte trans-endothelial migration, which once again proved the significant function of ECM remolding and immune inflammation in the disease process (*Figure 2H*).

## Dynamic transcriptional changes in ligamental degeneration

To gain insight into the cellular progression of fibroblast subclusters during the disease process, we performed cell trajectory analysis, revealing two significant routes in the process of degeneration. Five fibroblast subclusters, including Fib.2, Fib.4, Fib.5, Fib.9, and Fib.10, were involved in the reconstruction of the disease trajectories using Monocle 3, an algorithm for the reconstruction of lineage programs based on similarity at the transcriptional level (*Cao et al., 2019*). We set Fib.4 and Fib.5 as the starting point of the trajectories because they are structural fibroblasts with high expression of ECM component genes and represent normal ligamental functions well, and then computed pseudotime for cells along the inferred developmental axis (*Figure 3A and B*). More specifically, Fib.4 and Fib.5 were predicted to transform into two distinct cell fates, including the cell fate1, which includes Fib.2 and Fib.9, and the cell fate2, which includes Fib.10. With this in mind, we tried to explore the genic dynamics that distinguished these two cell fates. The expression profile of cell fate1 showed increasingly high expression of genes (CXCL12, MMP2, MMP14, C7) related to 'leukocyte trans-endothelial migration', 'cellular response to chemokine', and 'inflammatory response pathway'. Along with the cell fate2, we observed gradually high expression of ECM organization and skeletal muscle tissue development-related genes (FOS, EGR1, FGF10, FN1; *Figure 3C–H*).

To investigate gene expression dynamics along the trajectories, we group genes that varied between cell clusters into Module 10 using Louvain community analysis. We can observe the aggregated expression of each module in *Figure 3I*. We identified that the expression of genes in module 3 was gradually elevated along the cell fate1 and gradually declined along the cell fate2, which were enriched for genes related to rheumatoid arthritis, antigen processing and presentation, and complement activation. In contrast, the expression of wound repair and ECM assembly-related genes were gradually elevated along the cell fate2 but declined along the cell fate1, such as ADAM12, HS3ST3A1, FN1, and SERPINE2 in module 5 (*Figure 3I–K*). According to these pseudotime analysis results, we inferred that there are two opposite dynamic trajectories in the process of ligamental degeneration: one is the progressive damage process of complement inflammation leading to the continuous progression of the disease, and the other is the process of repairing the damage to delay or even reverse the disease process.

## Characterization of stromal cells in coordinating ligament microenvironment during degeneration progression

Besides fibroblasts, various endothelial cells (ECs) and immune cells also play significant roles in the occurrence and development of ligament degeneration. In our ligament samples, we identified seven clusters of blood-vessel-derived cells, including five clusters of ECs (EC1-5) and two clusters of pericytes (pericyte1 and pericyte2). *Figure 4A* shown the distribution of EC1 expressing a high level of ACKR1 is a population of venous ECs; EC2 highly expressed COL4A1, COL4A2, H19 was related to anti-angiogenesis; EC3 highly expressed genes associated with ECM organization and we inferred this cluster related to damage repair. EC4 expressing a high level of CCL2, CCL8, STEAP4, and SYNPO2 is a cluster of ECs related to inflammatory chemotaxis. EC5 is a cluster of lymphatic ECs. As for pericytes, we identified two populations that expressed pericyte markers ACTA2, and MCAM. Pericyte1 highly expressed myocyte-related genes such as MYH11, MUSTN1, and LBH, which implied that this cluster may have myocyte-like properties. Pericyte2 highly expressed fibroblast markers such as COL1A1

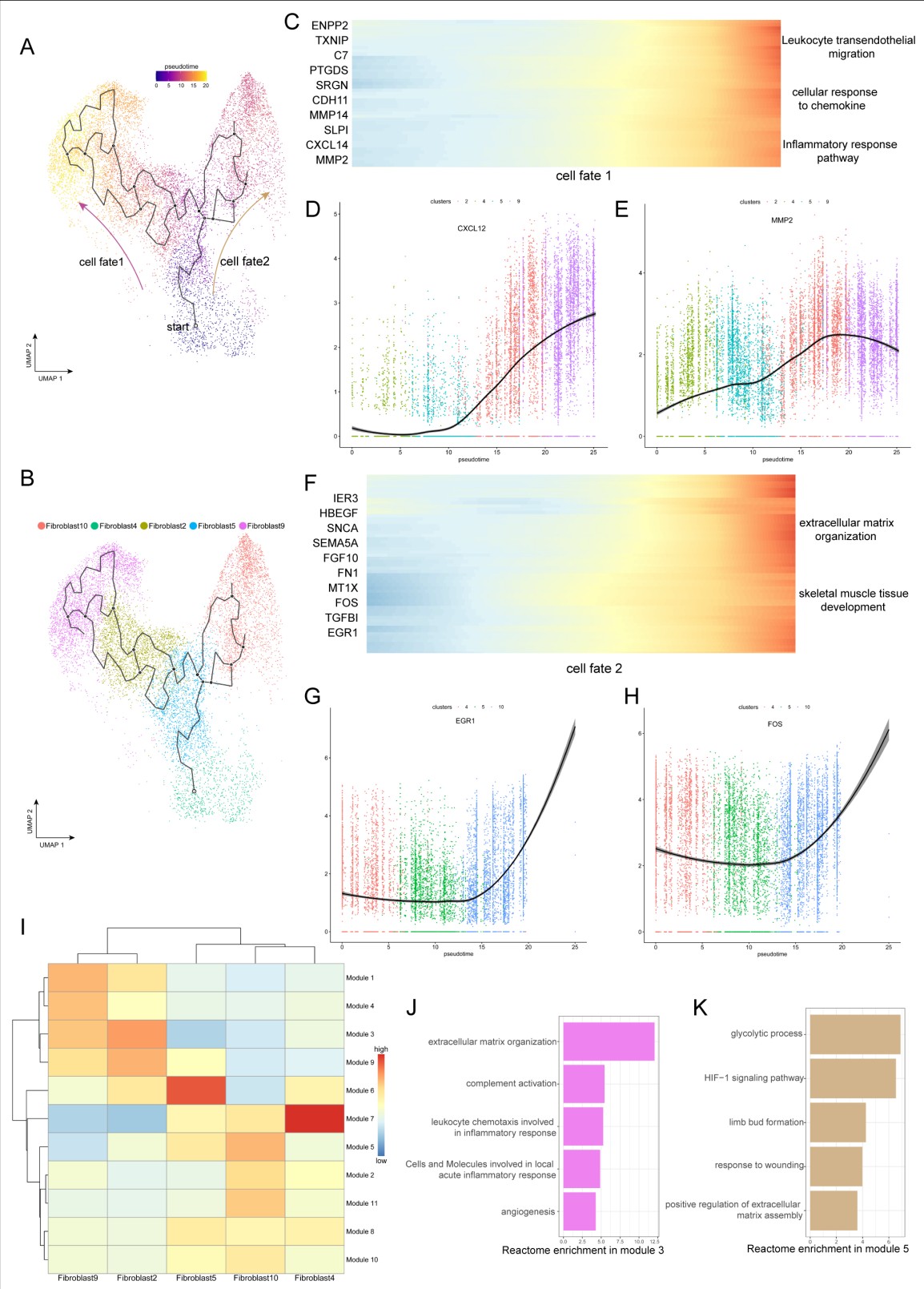

**Figure 3.** Evolution trajectory and transcriptional fluctuation during ligamental degeneration progression. (**A and B**) UMAP visualization of fibroblast 2, 4, 5, 9, 10. Developmental pseudotime for cells present along the trajectory inferred by Monocle 3, with cell fate1 and cell fate2 branches coming from fibroblast 4. (**C**) Heatmap showing the expression changes of the highly variable genes along the cell fate1 in normal and degenerated groups. (**D and E**) Representative gene expression levels along cell fate1 trajectory of normal and diseased statuses. (**F**) Heatmap showing the expression changes

*Figure 3 continued on next page*

Figure 3 continued

of the highly variable genes along the cell fate2 in normal and degenerated groups. (**G and H**) Representative gene expression levels along cell fate2 trajectory of normal and diseased statuses. (**I**) Heatmap showing the scaled mean expression of modules of coregulated genes grouped by Louvain community analysis across the subclusters. (**J and K**) Enrichment analysis results of Module 3 and 5.

and COL3A1, which implied that this cluster may have fibroblast-like properties (*Figure 4G*). We next compared the proportions of these 7 clusters between normal and diseased ligaments (*Figure 4Cand D*). The results illustrated that most ECs subsets were elevated in the diseased group, implying increased angiogenesis during ligamental degeneration. The ratio of pericyte1 in the diseased group was higher than that in the healthy group, but the ratio of pericyte2 was lower. An increased ratio of pericyte1 may change the tissue's biological properties, as ligament was dominant by fibroblast. To gain more biological insights underlying ECs, we performed the Degs analysis between normal and degenerated groups (*Figure 4E*). Combined with the result of enrichment analysis and GSEA analysis, we identified that ligamental ECs upregulated inflammation and complement-related genes, such as C1QA, PLA2G2A, and ECM-related genes such as PRG4, VIM, and POSTN during degeneration procession (*Figure 4H*). The terms 'inflammation mediated by chemokine and cytokine signaling pathway' and 'TGF-beta signaling pathway' activated in the diseased group also implied that inflammation and ECM remodeling have significant roles in the disease process (*Figure 4F*). By immunofluorescence staining, we demonstrated the presence of two pericyte subsets in normal and diseased samples. In the normal ligament, there were more pericyte2 than pericyte1 and compared with the normal group, pericyte1 was increased in the diseased group (*Figure 4I and J* and *Figure 4—figure supplement 1*).

As an indispensable cellular component in the ligament, we identified six subpopulations of immune cells, including basophils, CD8[+] NKT-like cells, macrophages, memory CD4[+] T cells, memory CD8[+] T cells, and myeloid dendritic cells (*Figure 4K*). From *Figure 4L and M*, we can conclude that macrophages are the most abundant immune cells in both normal and diseased groups and compared with the normal group, only the proportion of macrophages was increased in the degenerated group. We explored the gene profiling change of macrophages in the process of degeneration. As is shown in the violin plot *Figure 4N*, we suggested that ECM remolding-related genes have higher expression levels in the diseased group, such as PRG4, FN1, and HTRA1. The enrichment analysis also implied that ECM organization and complement and coagulation cascades were activated in the degenerated ligament, which was consistent with the results of ECs analysis (*Figure 4O*).

## Putative signaling network for the intercellular crosstalk regulating the microenvironment homeostasis during ligamental degeneration

Elucidating the explicit interaction among fibroblasts, ECs, and immune cells in the ligamental microenvironment will shed light on the mechanisms of the pathogenesis of ligamental degeneration. CellphoneDB and CellChat analysis were used to investigate the signaling network among the main cell clusters in the normal and diseased ligaments. The heatmap illustrated the intensity of interactions among different cell clusters. From the results, we can find that in addition to interactions within fibroblast subclusters, cell-cell cross-talks were mainly between fibroblast subpopulations and ECs and between fibroblast subpopulations and macrophages. At the same time, we can find that fibroblast subsets dominated by the diseased group had stronger interaction associations with other cells, which implied that there are stronger and more complex interactions in degenerated ligaments (*Figure 5A*).

To determine the important factors, we further analyzed the intercellular signaling networks of FGF and TGFB. Interestingly, Fib.9 was involved in FGF signaling, both autocrine and paracrine (*Figure 5B*). Fibroblasts were the leading receiver of FGF signals, as expected. Through the violin plot, we identified that Fib.9 acts on FGFR1 located in each fibroblast subpopulation by expressing FGF7 and FGF signaling pathways, which can promote pathological proliferation of cells and participate in scar repair (*Figure 5D*). We inferred that the FGF signaling network may disrupt the homeostasis of fibroblasts in the ligaments and promote the progression of ligamental degeneration. Moreover, the TGF-β pathway was involved in many cell-cell interactions among fibroblast subpopulations and macrophages via TGFB1-TGFBR1 or TGFB1-TGFBR2 (*Figure 5C and E*). As shown above, fibroblasts of the degenerated group highly expressed PRG4, COL3A1, and SSP1, and these genes are all the important downstream targets of TGF-β, which participate in the process of ECM remodeling. Hence,

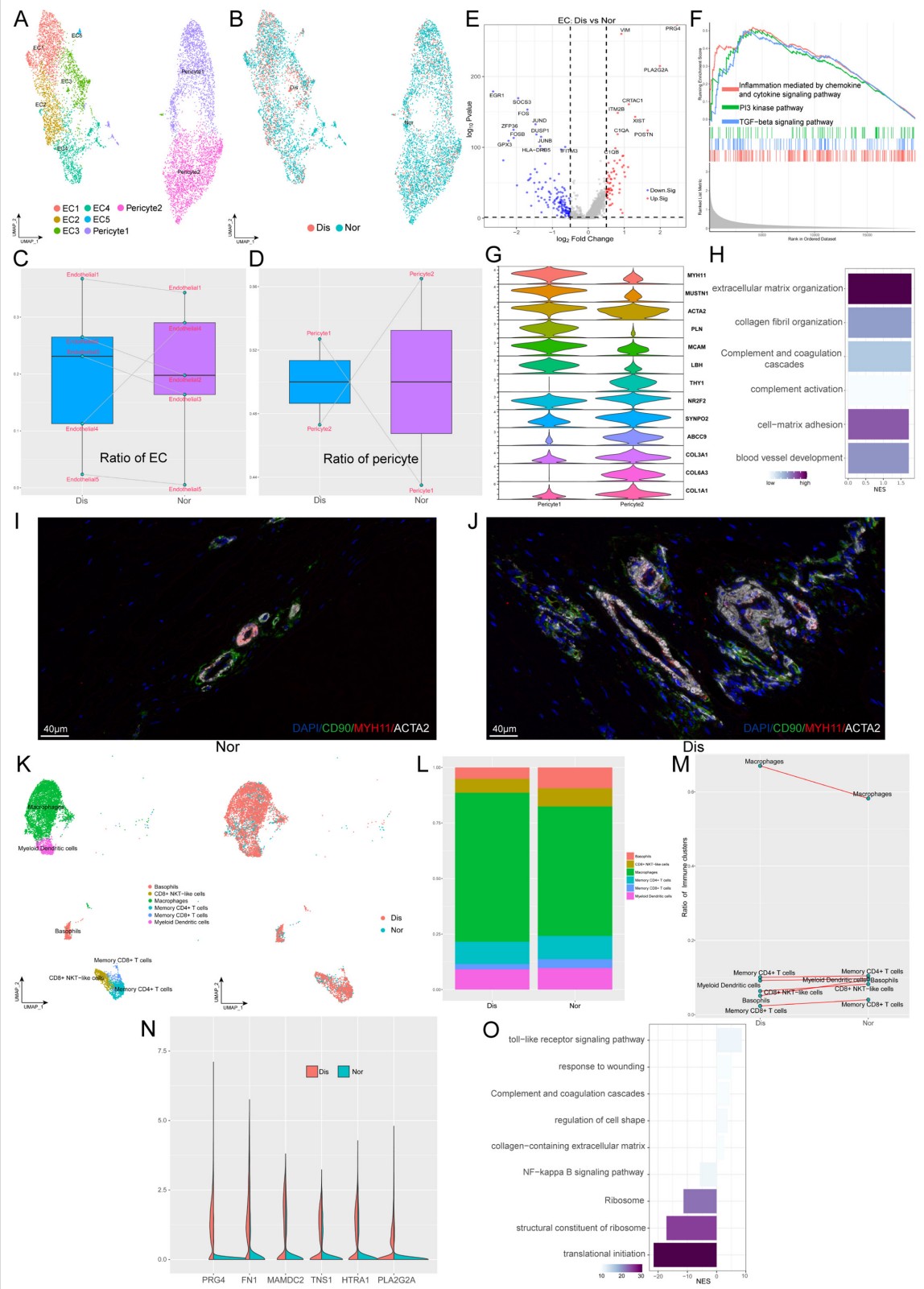

**Figure 4.** Identification of blood vessel derived cell and immune cell subclusters in human ligament. (**A**) UMAP visualization of the subclusters of endothelial cell and pericyte. (**B**) UMAP visualization of the distribution of endothelial cell and pericyte subclusters at different sample statuses. (**C and D**) Summarized subpopulations of endothelial cell and pericyte percentage changes. (**E**) Volcano plots displaying the DEGs in endothelial cell between normal group and diseased group. Each dot represented one gene. Red dots, differentially up-regulated genes; blue dots, differentially down-regulated

*Figure 4 continued on next page*

*Figure 4 continued*

genes; gray dot, non-differentially expressed genes. (**F**) GSEA enrichment plots for representative signaling pathways upregulated in endothelial cell of diseased samples, compared with normal samples. (**G**) Violin plots showing representative marker genes associated with different types of pericyte expressed in pericyte1 and pericyte2. (**H**) GO and KEGG enrichment analysis of DEGs between normal and diseased endothelial cells. (**I and J**) Immunofluorescence staining of pericyte related markers in normal and diseased groups. (**K**) UMAP visualization of the subclusters of immune cell and the distribution of immune cell subclusters at different sample statuses. (**L**) The proportion of each subcluster of immune cells in the lesioned and normal ligament. (**M**) Summarized subpopulations of immune cell percentage changes. (**N**) Violin plots showing representative genes of macrophages between normal and degenerated states. (**O**) GO and KEGG enrichment analysis of DEGs between normal and diseased macrophages.

The online version of this article includes the following figure supplement(s) for figure 4:

**Figure supplement 1.** Immunofluorescence staining of ACTA2, CD90, and MYH11 in normal and degenerated ACLs.

we suggested that TGFB1 is a key gene and Fib.8 and macrophages promote the development of the disease through this molecule. According to the Degs, we selected APOE as the marker gene of Fib.9 and TOP2A as the marker gene of Fib.8. The results of immunofluorescent staining illustrated that the number of Fib.8 and Fib.9 in the disease group were more than those in the normal group, and Fib.8 highly expressed TGFB1 and Fib.9 highly expressed FGF7 in the disease group, which validated the above findings (*Figure 5F* and *Figure 5—figure supplement 1*).

## SpRNA-seq of ligament deciphering the spatial interactions of cell subclusters

For further insight, and multi-angle interpretation of the cell composition changes that occurred in ligamental degeneration, we performed spRNA-seq on normal and lesioned ligaments. Firstly, by using the feature marker transfer method, we mapped the individual subsets identified by single-cell sequencing into spatial transcriptome data. The heatmap illustrated that Fib.4, as a representative of fibroblasts in the normal ligament tissue, was widely distributed in L1 and the amount was significantly higher than that in L8. The number of Fib.9, as a representative of fibroblasts in the degenerated ligament, in L8 was higher than that in L1 (*Figure 6A–D*). We also examined the immune cells and ECs. As expected, there were more immune cells and ECs in L8 than in L1, implying immune infiltration and vascular hyperplasia were presented in the degenerative specimens (*Figure 6E–J*). In addition, we also examined the FGF and TGFB signaling pathways related genes expression in spRNA-seq. From the results, we can find that TGFB1, TGFBR1, FGF7, and FGFR1 were highly expressed in the diseased sample (*Figure 6—figure supplement 1*). These results are consistent with those obtained by scRNA-seq. After that, we performed spotlight analysis in these two groups. All cell subsets are rendered in four colors, with red being the fibroblast subsets specific to the disease group. Finally, we hide the areas that are not red in both samples. We can find that the red region in L8 was widely distributed, and the area was significantly higher than that in L1. After careful observation, we found that almost all the red dots had yellow and black areas (*Figure 6K and L*). This indicates that some fibroblast subpopulations specific to the diseased group are adjacent to endothelial and immune cells in the degenerative state. The proximity of different cells in space makes it possible for them to interact with each other. Therefore, this further provided strong evidence for the results of our single-cell interaction analysis.

## Discussion

ACL degeneration can contribute to cartilage injury and even OA onset and progression and may put a great burden on the normal life of many patients (*Georgoulis et al., 2010*; *Hasegawa et al., 2012*). However, the mechanisms of this disease are not well characterized and treatments to prevent or treat ACL degeneration are scarce and not effective (*Shane Anderson and Loeser, 2010*; *Tozer and Duprez, 2005*). Healthy and degenerated ACL tissues include multiple cell subpopulations with diverse genetic and phenotypic characteristics. How this heterogeneity emerges in development degeneration remains unclear. Herein, we built a single-cell atlas of normal and degenerated human ACL and explored the characteristics and key regulatory pathways of distinct fibroblast subtypes. These findings will help us understand the pathogenesis of ACL degeneration in-depth, and provide potential targets for clinical therapies for this disease.

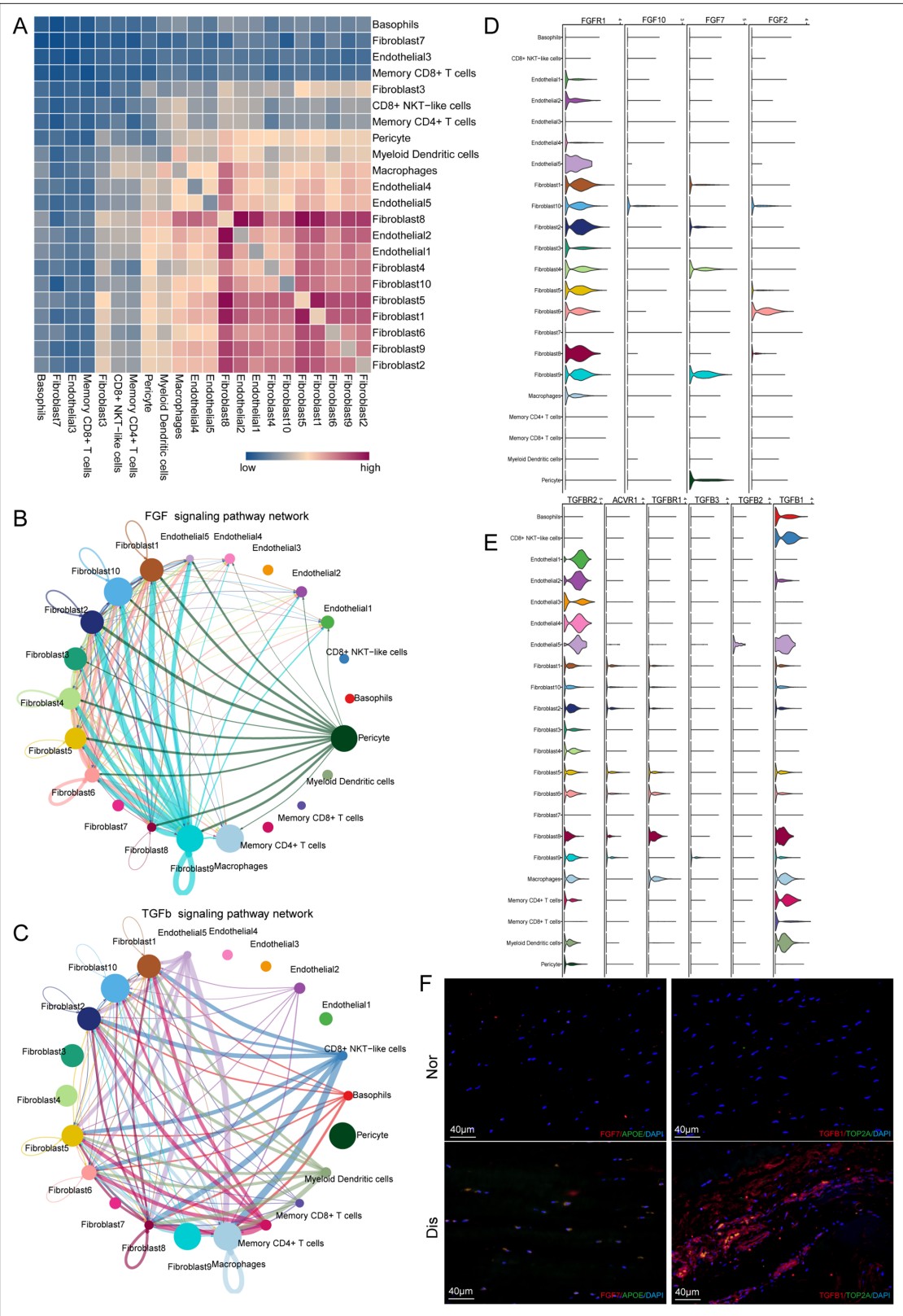

**Figure 5.** Cell–cell crosstalk during ligamental degeneration progression. (**A**) Heatmap depicting the significant interactions among the identified subclusters of fibroblast, immune cell, and endothelial cell of normal and degenerated ligaments. (**B and C**) Circle plots showing the inferred TGF-β and FGF signaling networks. (**D and E**) Violin plots showing ligand-receptor interactions related to TGF-β and FGF signaling pathways among subclusters of fibroblast, endothelial cell and immune cell. (**F**) Immunofluorescent staining of TGFβ1, TOP2A, APOE, and FGF7 in normal and diseased groups.

*Figure 5 continued on next page*

*Figure 5 continued*

The online version of this article includes the following figure supplement(s) for figure 5:

**Figure supplement 1.** Immunofluorescence staining of FGF7, APOE, TGFB1, and TOP2A in normal and degenerated ACLs.

Fibroblasts are increasingly considered dominating cell types in ligaments and central mediators of ligamental degeneration, and here, we identified 10 fibroblast subpopulations in normal and degenerated ACL samples by using scRNA-seq. Further cluster analysis profiled characteristics of each subcluster and combined with the cell proportion analysis, we found that pro-inflammation and ECM remodeling-related fibroblast subgroups were significantly increased in the degenerative ligamental tissues, which implied that inflammation and ECM remodeling are key events in the disease process. Through comparing the Degs of fibroblasts between degenerative and healthy conditions, we found that ECM-related genes upregulated in the diseased group, such as FN1, COL3A1, PRG4, and SPP1. The results of enrichment analysis and GSEA also illustrated that ECM-related pathways were activated in the degenerated group, which suggested that the modules of ECM play an important role in the process of ligamental degeneration. According to the gene interaction analysis, two gene modules were activated in the diseased group, leukocyte trans-endothelial migration and ECM-receptor interaction. These findings were consistent with previous studies which suggested that ligament inflammation and ECM changes contribute to ACL degeneration (*Busch et al., 2013*; *Hasegawa et al., 2012*). It is due to the changes and remodeling of the ECM in the degenerative ligament tissue that the biomechanical changes of the ligament are caused and eventually may result in the rupture of the ligament. From the pseudotime analysis, we identified two cell fates in the ligamental degeneration. Cell fate 1 represents a progressive process of the disease characterized by inflammatory damage and extracellular matrix degradation. Cell fate 2 implies a chronic repair process for the disease characterized by ECM remodeling, ligament development, and damage repair.

Immune cells are closely related to ligamental degeneration (*Kim-Wang et al., 2021*). We analyzed the heterogeneity of immune cells in ACL using single-cell RNA sequencing. The results illustrated that macrophage-dominating immune cells in ACL and inflammation-related genes PLA2G2A and ECM-related genes PRG4, FN1, and HTRA1 were upregulated in the macrophages of the diseased group. Enrichment analysis also revealed that complement and ECM-related pathways were activated in the degenerated group. This implied that macrophages highly expressed ECM and inflammation-associated genes in the ligamental degeneration progression. So, we suggested that macrophages may contribute to ligamental degeneration. We also identified two types of pericytes in these tissues. As we all know, pericytes in the tissue have some stem cell characteristics and can differentiate into fibroblasts, and chondrocytes (*Armulik et al., 2011*; *Smyth et al., 2018*). Pericyte 1 highly expressed stem cell-related genes MCAM and MUSTN1 and have the characteristics of myofibroblasts, and pericyte 2 tends to have the characteristics of fibroblasts. Pericyte 1 was increased and pericyte 2 decreased in the degenerated group, which may change the properties of ligamental cells and contribute to ligamental degeneration.

Cells can communicate via ligand-receptor interactions (*Kumar et al., 2018*), so targeting cell-cell interactions is frequently utilized in clinical treatment. The results of CellphoneDB analysis demonstrated that the high content of fibroblasts and immune cells in the degenerative group had significantly higher interaction intensity, such as fibroblast 1, 2, and 8, which means that the interaction between cells in the ligament was enhanced in the degenerative state. To identify the key factors regulating the disease process, CellChat analysis was used to dissect the intercellular crosstalk based on the signaling network in the human ACL. We found that the FGF signaling pathway and TGF-β signaling pathway were involved in the crosstalk network. TGFβ1 contributes to osteophyte formation by inducing endochondral ossification, is thought to be closely related to the onset of OA (*Murata et al., 2019*), and can be used to induce chondrogenic differentiation of ligament-derived stem cells in vitro (*Schwarz et al., 2019*). It has been reported that in myocardial fibrosis, liver fibrosis, glaucoma, and other diseases, the TGF-β signaling pathway is also involved in angiogenesis, ECM remodeling, pathological scar healing (*Ilieş et al., 2021*; *Lee and Massagué, 2022*; *Ferrão et al., 2018*; *Penn et al., 2012*). FGFs can induce the fibroblast to myofibroblast differentiation, promote tissue repair by regulating cell proliferation, survival, and angiogenesis and contribute to ECM remodeling (*Kendall and Feghali-Bostwick, 2014*; *Ma et al., 2017*). ECM remodeling by overexpression, degradation, and cross-linking of ECM proteins in the lesioned tissue are direct factors leading to fibrosis

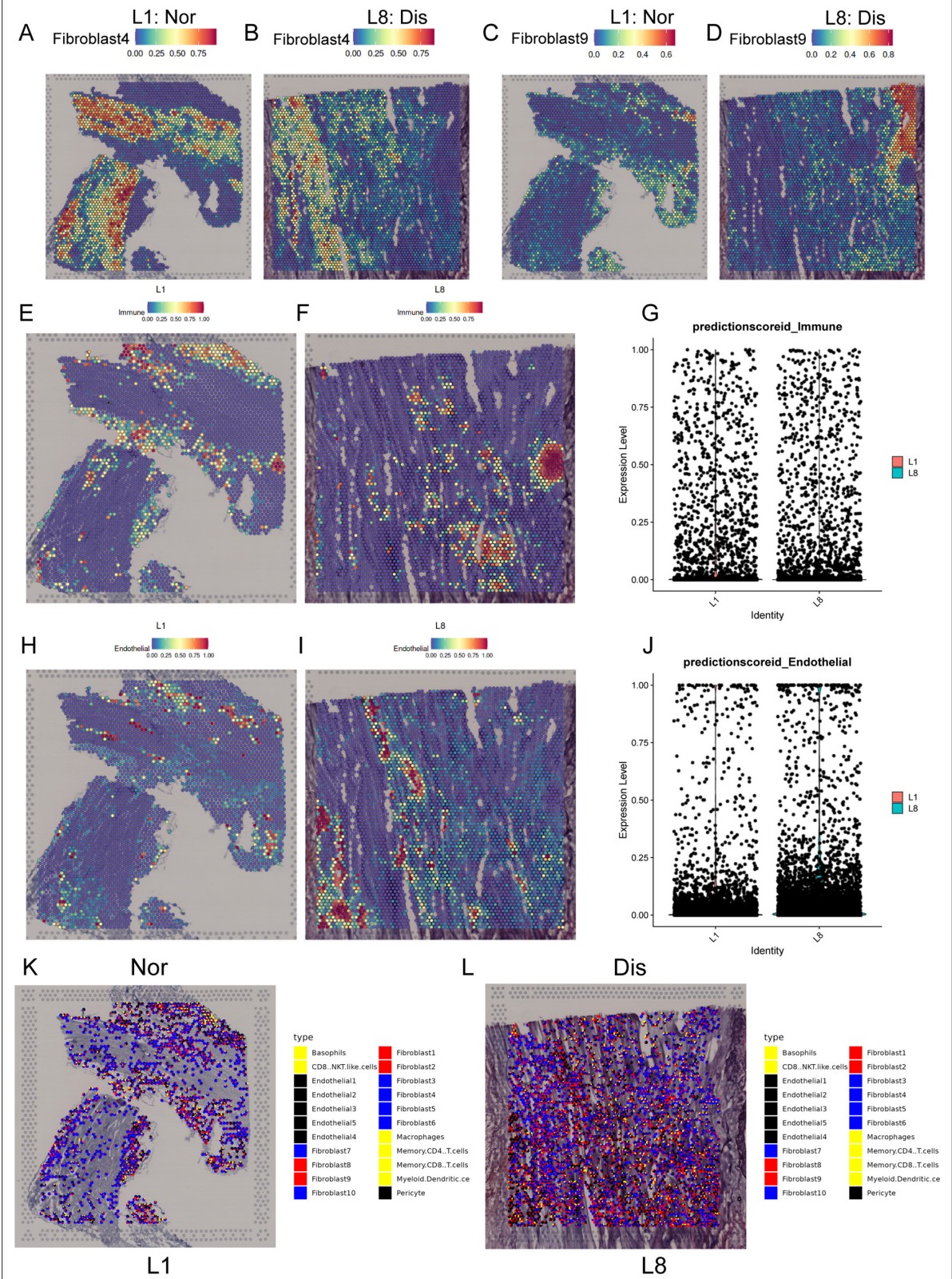

**Figure 6.** Spatial transcriptome sequencing deciphers the microenvironment changes during ligamental degeneration progression. (**A–D**) Spatial heatmaps showing the representative normal and degenerated fibroblast subcluster distribution in L1 and L8 samples. (**E and F**) Spatial heatmaps showing the immune cells distribution in normal and degenerated ligamental samples. (**G**) Quantitative analysis of immune cells in these two types of tissues. (**H and I**) Spatial heatmaps showing the endothelial cells distribution in normal and degenerated ligamental samples. (**J**) Quantitative analysis of

*Figure 6 continued on next page*

*Figure 6 continued*

endothelial cells in these two types of tissues. (**K and L**) Spot light showing fibroblast subpopulation unique to diseased samples proportion in L1 and L8 samples.

The online version of this article includes the following figure supplement(s) for figure 6:

**Figure supplement 1.** Expression data of FGF7 and TGFβ ligand and receptor genes in the spatial transcriptomes.

tissues (*Sun et al., 2022*). In this study, we observed that fibroblast9, inflammation-related fibroblast, highly expressed FGF7 and can act on FGFR1 mainly distributed in the fibroblast subgroups. Immune cells especially macrophages highly expressed TGFB1 and can act on TGFBR1 and TGFBR2 mainly distributed in the fibroblast and endothelial subgroups. These findings suggested that FGF and TGF-β signaling pathways may induce ECM remolding in the process of ligamental degeneration and FGF7-FGFR1 and TGFB1-TGFBR2 may be potential targets for the treatment of this disease. SpRNA-seq was used to detect the spatial information underlying the normal and degenerated ligaments and validate the findings acquired from the scRNA-seq. Through the results, we observed that the disease group specific fibroblasts and immune cells recognized by single cells were more numerous and more widely distributed in the disease group specimens. We also got some spatial information, that fibroblasts in the disease group were spatially closer to immune and endothelial cells, which was more conducive to cell interaction.

## Conclusions

In conclusion, our study described the cell atlas of the human ligament, providing a valuable resource for further investigation of ACL homeostasis and the pathogenesis of ligamental degeneration. The cellular heterogeneity and signaling network we uncovered help to increase the understanding of the human ACL at a single-cell level and provide crucial clues for establishing new diagnostic and therapeutic strategies for this disease in the future.

## Acknowledgements

We appreciate all patients who participated in this study. The authors are grateful to Yi Zhang, Yue Li, and Wanli Zhang for the help in multiplex immunofluorescence staining. We would like to thank Genergy Bio-technology who provided scRNA-seq and spRNA-seq support for this work.

## Additional information

### Funding

| Funder | Grant reference number | Author |
|---|---|---|
| National Natural Science Foundation of China | 81972123 | Weili Fu |
| National Natural Science Foundation of China | 82172508 | Weili Fu |
| National Natural Science Foundation of China (82372490) | | Weili Fu |
| 1.3.5 Project for Disciplines of Excellence of West China Hospital Sichuan University (ZYJC21030, ZY2017301) | | Weili Fu |

The funders had no role in study design, data collection and interpretation, or the decision to submit the work for publication.

## Author contributions
Runze Yang, Resources, Software, Validation, Methodology, Writing - original draft, Project administration; Tianhao Xu, Software, Supervision, Validation, Visualization; Lei Zhang, Minghao Ge, Supervision, Validation, Visualization; Liwei Yan, Software, Supervision, Methodology; Jian Li, Conceptualization, Resources, Data curation, Formal analysis, Supervision; Weili Fu, Conceptualization, Data curation, Formal analysis, Supervision, Funding acquisition, Investigation, Writing - original draft, Project administration, Writing - review and editing

## Author ORCIDs
Weili Fu http://orcid.org/0000-0003-4438-2760

## Ethics
This study was reviewed and approved by our University Ethics Committee (Ethics Committee on Biomedical Research, West China Hospital of Sichuan University No. 658 2020-(921)) and all procedures complied with the Helsinki Declaration. Participants gave informed consent to participate in the study.

## Decision letter and Author response
Decision letter https://doi.org/10.7554/eLife.85700.sa1
Author response https://doi.org/10.7554/eLife.85700.sa2

# Additional files

## Supplementary files
• Supplementary file 1. Donor and sample information.
• MDAR checklist

## Data availability
Data are available in a public, open-access repository. The single-cell RNA-seq data and cluster annotations are available at GSA for human (https://ngdc.cncb.ac.cn/gsa-human/) with the accession number PRJCA014157.

The following dataset was generated:

| Author(s) | Year | Dataset title | Dataset URL | Database and Identifier |
| --- | --- | --- | --- | --- |
| Fu W | 2023 | The single-cell RNA-seq data and cluster annotations are available at GSA for human | https://ngdc.cncb.ac.cn/bioproject/browse/PRJCA014157 | CNCB BioProject, PRJCA014157 |

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
