## [Editor Report]

This is a valuable study of particular interest to those researchers working in the field. The authors provide a single-cell atlas of healthy and degenerative human anterior cruciate ligament using transcriptomic profiling. The quality of the study was considered valuable as it expands current knowledge.

---

## [Decision Letter]

**Decision letter after peer review:**

Thank you for submitting your article "A single-cell atlas depicting the cellular and molecular features in human ligamental degeneration: a single cell combined spatial transcriptomics study" for consideration by *eLife*. Your article has been reviewed by 3 peer reviewers, and the evaluation has been overseen by a Reviewing Editor and Carlos Isales as the Senior Editor. The reviewers have opted to remain anonymous.

Essential revisions:

1) The initial findings provided by the scRNA-seq technique need to be validated by in vitro and in vivo experiments.

2) The "Introduction" section could be improved substantially by removing irrelevant and redundant information. This section should focus on the biology of ACL, particularly current knowledge about the cellular and molecular mechanisms that maintain ACL homeostasis or cause degeneration.

3) The legend of Figure S1 D-G and the corresponding text needs to be revised to increase clarity. The current layouts are confusing, and those violin plots seem to present scRNA-seq data. There wasn't any comparison of gene expression between the bulk RNA-seq and scRNA-seq data per se.

In addition, the authors need to address other comments raised by 3 reviewers.

*Reviewer #1 (Recommendations for the authors):*

In this study, the authors performed single-cell RNA sequencing using normal and diseased human anterior cruciate ligaments. However, the study is very preliminary, which needs to be confirmed by animal studies and in vitro studies.

1) The authors identified two types of pericyte cells with different characteristics. If this can be reproduced by animal study, it will strengthen the findings reported by single-cell RNA sequencing assay.

2) Roles of TGF-β and FGF signaling in the anterior cruciate ligament degeneration need to be further verified by animal studies and by in vitro studies.

*Reviewer #2 (Recommendations for the authors):*

1) Some of the descriptions in the text are inappropriate. Such as L25 "two stages of the degenerative process". L224: "obviously higher than that in the healthy group".

2) In figure3, the author only kept 4 clusters for the pseudotime analysis, what's the reason for excluding the other subpopulations?

3) The grammar throughout the manuscript needs significant improvement.

*Reviewer #3 (Recommendations for the authors):*

Concerns:

1. The "Introduction" section could be improved substantially by removing irrelevant and redundant information. This section should focus on the biology of ACL, particularly current knowledge about the cellular and molecular mechanisms that maintain ACL homeostasis or cause degeneration.

2. The authors should check and review extensively for improvements to the use of English.

3. The labels in Figure S1C-G used to define the normal and disease groups and cell clusters (fibroblasts, immune cells, pericytes, and endothelial cells) are too small to read. In Figure 1F, the genes listed on the left didn't match the corresponding heatmap, and their color codes were not defined.

4. Page 10, lines 204-206 and Figure S1 D-G: these figures, figure legends, and corresponding text need to be revised to increase clarity. The current layouts are confusing, and those violin plots seem to present scRNA-seq data. There wasn't any comparison of gene expression between the bulk RNA-seq and scRNA-seq data per se.

5. Page 10, lines 208-209: Please consider revising it as "Characterization of fibroblast subpopulations in healthy and degenerative ACL". The rationale and criteria used for classifying subclusters of the ACL fibroblasts are ambiguous (Figures 2&3). This classification lacks the support of functional validation, therefore dampening its authenticity and significance.

6. The Data shown in Figure 2 F-H are computational models; they need to be validated by additional experiments. This applies to other claimed findings as well.

7. Considering the effect of biological variables and age on this study, information about the donors' gender and age should be provided.

[Editors' note: further revisions were suggested prior to acceptance, as described below.]

Thank you for resubmitting your work entitled "A single-cell atlas depicting the cellular and molecular features in human ligamental degeneration: a single cell combined spatial transcriptomics study" for further consideration by *eLife*. Your revised article has been evaluated by Carlos Isales (Senior Editor) and a Reviewing Editor.

The manuscript has been improved but there are some remaining issues that need to be addressed, as outlined below:

1. The authors are suggested to run the proofreading carefully since many typos are found in the text. Below is a list of some, but not all.

P3 line 48: consider changing " the musculoskeletal" to "the musculoskeletal system".

P3 line 59: Please consider revising "ACL aging and degradation" to "ACL aging and degeneration". Also, "degradation" has been heavily used in the text. Consider replacing it with "degeneration" wherever it is appropriate.

P4 line 85: replace "synergize" with "synergy"; P10 line 204: please replace "resolve" with "reveal"?

2. The color codes for Figure 2B are confusing, please consider changing the color codes for L3 to L5 to increase visibility.

3. Markers used for pericyte immunostaining in Figure 4G were incorrect! ACTA2, MYH11, and CD90 (Thy1) are not specific markers for pericytes. ACTA2 (also called sSMA) and MYH11 (myosin11) are generally considered markers for myofibroblasts and vascular smooth muscle, not pericytes. Also, most of the mesenchymal progenitors express CD19. So pericyte classification in this ms is confusing and not supported by the immunostaining data.

4. The data in Figure 5F doesn't convincingly demonstrate there is an increase in the expression of FGF7 and TGFβ1 (where and which types of cells) in the specimens from the diseased groups. In addition, these data need to be quantified, and the "n" number should be provided.

---

## [Author Response]

Essential revisions:1) The initial findings provided by the scRNA-seq technique need to be validated by in vitro and in vivo experiments.

Thanks for the comments and suggestions. We followed them and mainly validated the changes of fibroblasts during ligamental degeneration and the changes of two pericyte subsets in normal and degenerative specimens by immunohistochemical staining, and immunofluorescence staining. In addition, two disease-related pathways identified by the scRNA-seq: TGF-β and FGF signaling were verified by immunohistochemical staining and validated at the spatial transcriptome level.

2) The "Introduction" section could be improved substantially by removing irrelevant and redundant information. This section should focus on the biology of ACL, particularly current knowledge about the cellular and molecular mechanisms that maintain ACL homeostasis or cause degeneration.

Thank you for your suggestions. We have reworded the “Introduction” section according to your requirement. The revised “Introduction” section focuses on the current knowledge about the cellular and molecular mechanisms that maintain ACL homeostasis or cause degeneration.

3) The legend of Figure S1 D-G and the corresponding text needs to be revised to increase clarity. The current layouts are confusing, and those violin plots seem to present scRNA-seq data. There wasn't any comparison of gene expression between the bulk RNA-seq and scRNA-seq data per se.

Sorry for the inconvenience. We have now re-written the legend of Figure S1 D-G and the corresponding text. These violin plots do not represent scRNA-seq data. Here, we make a detailed explanation as follows. We first selected genes that were significant highly expressed in the diseased group than in the normal group in the four clusters (fibroblast, endothelial cell, immune cell, and pericyte) of our scRNA-seq data, and then we detected these genes in the bulk RNA-seq data to verify whether these genes were also highly expressed in the diseased group of the bulk RNA-seq. Thus, the consistency of our scRNA-seq data with the bulk RNA-seq data was demonstrated by these violin plots. In addition, according to your suggestions, we added the direct comparison of gene expression between the bulk RNA-seq and scRNA-seq data per se.

Reviewer #1 (Recommendations for the authors):In this study, the authors performed single-cell RNA sequencing using normal and diseased human anterior cruciate ligaments. However, the study is very preliminary, which needs to be confirmed by animal studies and in vitro studies.1) The authors identified two types of pericyte cells with different characteristics. If this can be reproduced by animal study, it will strengthen the findings reported by single-cell RNA sequencing assay.

Thanks. According to your suggestions, we selected the marker genes of two types of pericytes and performed immunofluorescence staining in normal and diseased samples, which confirmed the findings reported by scRNA-seq.

2) Roles of TGF-β and FGF signaling in the anterior cruciate ligament degeneration need to be further verified by animal studies and by in vitro studies.

Thanks for your suggestions. We verified the expression of key genes in these two signaling pathways: FGF7 and TGFB1 in the normal and diseased groups by immunohistochemical staining, and also detected these two pathways at the spatial transcriptome level. As a single-cell atlas type article, we aim to construct single-cell maps of both normal and degenerative ligaments to provide tools and resources for future research in ligament-related diseases. So relatively little experimental verification has been done.

Reviewer #2 (Recommendations for the authors):1) Some of the descriptions in the text are inappropriate. Such as L25 "two stages of the degenerative process". L224: "obviously higher than that in the healthy group".

Sorry for that. According to your suggestions, we have reworded the inappropriate descriptions in the manuscript.

2) In figure3, the author only kept 4 clusters for the pseudotime analysis, what's the reason for excluding the other subpopulations?

Thanks for your great questions. We kept 5 clusters (fibroblast2,4,5,9,10) for the pseodotime analysis. According to the Degs and the definition of different fibroblast subclusters, these five cell populations are the focus of our attention and are closely related to the disease progression. Besides, based on the current understanding of ligamental degeneration, we hypothesized that the process of ligamental degeneration could be reflected in these cell clusters by the pseodotime analysis. In addition, the reasons for excluding the other subpopulations are also described. Fibroblast8: a subpopulation that are in the cell cycle is not associated with disease progression. Fibroblast7: a subpopulation that has not yet been defined. Fibroblast6: a cluster of chondrocytes and not associated with ligamental degeneration. Fibroblast3: high expression of metalloenzyme related genes and low correlation with the disease. Fibroblast1: a subpopulation associated with ECM remodeling. It was initially included in the pseodotime analysis, but the results did not fit the biological process, so it was also excluded.

3) The grammar throughout the manuscript needs significant improvement.

Thanks for your comments and suggestions. We carefully checked and revised the grammar in the manuscript. Besides, we also asked for help from the native speaker on the grammar and syntax issues.

Reviewer #3 (Recommendations for the authors):Concerns:1. The "Introduction" section could be improved substantially by removing irrelevant and redundant information. This section should focus on the biology of ACL, particularly current knowledge about the cellular and molecular mechanisms that maintain ACL homeostasis or cause degeneration.

Thanks. According to your suggestions, we have reworded the “Introduction” section according to your requirement. The revised “Introduction” section focuses on the current knowledge about the cellular and molecular mechanisms that maintain ACL homeostasis or cause degeneration.

2. The authors should check and review extensively for improvements to the use of English.

Thanks for your reminding. We have carefully checked and reviewed the use of English of the manuscript and ask for help from the native speaker. Besides, we have extensively and rigorously revised the English use in the manuscript.

3. The labels in Figure S1C-G used to define the normal and disease groups and cell clusters (fibroblasts, immune cells, pericytes, and endothelial cells) are too small to read. In Figure 1F, the genes listed on the left didn't match the corresponding heatmap, and their color codes were not defined.

Thanks for your suggestions. We adjusted the labels in Figure S1C-G and redrew the Figure 1F.

4. Page 10, lines 204-206 and Figure S1 D-G: these figures, figure legends, and corresponding text need to be revised to increase clarity. The current layouts are confusing, and those violin plots seem to present scRNA-seq data. There wasn't any comparison of gene expression between the bulk RNA-seq and scRNA-seq data per se.

Sorry for the inconvenience. We have now re-written the legend of Figure S1 D-G and the corresponding text. These violin plots do not represent scRNA-seq data. Here, we make a detailed explanation as follows. We first selected genes that were significant highly expressed in the diseased group than in the normal group in the four clusters (fibroblast, endothelial cell, immune cell, and pericyte) of our scRNA-seq data, and then we detected these genes in the bulk RNA-seq data to verify whether these genes were also highly expressed in the diseased group of the bulk RNA-seq. Thus, the consistency of our scRNA-seq data with the bulk RNA-seq data was demonstrated by these violin plots.

In addition, according to your suggestions, we added the direct comparison of gene expression between the bulk RNA-seq and scRNA-seq data per se and a GSVA analysis. Through the results of direct comparison and GSVA analysis (figure 1L and supplementary figure 1C), we further proved the reliability of our scRNA-seq results.

5. Page 10, lines 208-209: Please consider revising it as "Characterization of fibroblast subpopulations in healthy and degenerative ACL". The rationale and criteria used for classifying subclusters of the ACL fibroblasts are ambiguous (Figures 2&3). This classification lacks the support of functional validation, therefore dampening its authenticity and significance.

Thank you for your suggestions. For lines 208-209, We have revised it as “Characterization of fibroblast subpopulations in healthy and degenerative ACL”. We believed that our classification of fibroblast subclusters are objective and rational. We defined them based on the characteristic Degs of each subclusters and combined the function of multiple genes to predict the function of each fibroblast subcluster. At the same time, we added the GSVA method to validate our fibroblast classification criteria in bulk RNA-seq samples, and the results further proved the rationality of our classification.

6. The Data shown in Figure 2 F-H are computational models; they need to be validated by additional experiments. This applies to other claimed findings as well.

Thanks. Immunohistochemical and immunofluorescence staining of normal and disease ligaments were used to validate our results. At the same time, the results of spatial transcriptome sequencing also confirmed the results of scRNA-seq.

7. Considering the effect of biological variables and age on this study, information about the donors' gender and age should be provided.

Thank you for your reminding. We added the information about the donors’ gender and age in the supplementary table.

[Editors’ note: what follows is the authors’ response to the second round of review.]

The manuscript has been improved but there are some remaining issues that need to be addressed, as outlined below:1. The authors are suggested to run the proofreading carefully since many typos are found in the text. Below is a list of some, but not all.P3 line 48: consider changing " the musculoskeletal" to "the musculoskeletal system".P3 line 59: Please consider revising "ACL aging and degradation" to "ACL aging and degeneration". Also, "degradation" has been heavily used in the text. Consider replacing it with "degeneration" wherever it is appropriate.P4 line 85: replace "synergize" with "synergy"; P10 line 204: please replace "resolve" with "reveal"?

Thanks for your suggestions. We have carefully proofread and revised the whole manuscript.

2. The color codes for Figure 2B are confusing, please consider changing the color codes for L3 to L5 to increase visibility.

Thanks. According to your suggestions, we have changed the color codes in Figure 2B to make the figure clearer.

3. Markers used for pericyte immunostaining in Figure 4G were incorrect! ACTA2, MYH11, and CD90 (Thy1) are not specific markers for pericytes. ACTA2 (also called sSMA) and MYH11 (myosin11) are generally considered markers for myofibroblasts and vascular smooth muscle, not pericytes. Also, most of the mesenchymal progenitors express CD19. So pericyte classification in this ms is confusing and not supported by the immunostaining data.

Thank you for the suggestion. Our definition of pericytes is well-founded and markers we selected used for pericyte immunostaining are also reasonable. Firstly, pericytes and smooth muscle cells originate from the same mesenchymal lineage, share many marker expressions like PDGFRB, ACTA2, CD13, CD146^1^. Pericytes also present strong muscle contractile gene expression signatures, including ACTA2, MYH/MYL genes, TAGLN^2^. There are several literature demonstrating pericytes expressing ACTA2 and MYH/MYL genes across multiple human organs such as meniscus^2^, intervertebral disc^3^, heart^4^, brain^5^. Secondly, it is known that pericytes attach to the surface of vascular endothelial cells and maintain vasculature stability via the endothelium–pericyte crosstalk^6^. Our immunofluorescence imaging found that this population of cells formed a tube shape and it seems to surround the blood vessels. Therefore, according to the spatial localization, this group of cells also conforms to the characteristics of pericytes. Thirdly, it has been reported that there are two subsets of pericytes. One subset expresses more stem cell markers such as CD90, RGS5, CD146. Another subset expressed more markers associated with muscle contraction, such as MYH11, MYH9^1, 2^. In our study, we also found similar two populations of pericytes and performed multiple immunofluorescent staining according to their markers (ACTA2: shared marker; MYH11: a marker of pericyte1; THY1/CD90: a marker of pericyte2). So, we believe that pericyte classification in our study is clear and can be supported by our immunostaining data.

4. The data in Figure 5F doesn't convincingly demonstrate there is an increase in the expression of FGF7 and TGFβ1 (where and which types of cells) in the specimens from the diseased groups. In addition, these data need to be quantified, and the "n" number should be provided.

Thank you for your suggestions. Fibroblasts are the main cells in the ligament and are the focus of our study. In order to better reflect which types of cells express TGFB1 and FGF7, we conducted immunofluorescence co-staining of Fib.9 marker gene APOE and FGF7 and Fib.8 marker gene TOP2A and TGFB1 according to the results of cell interaction analysis. According to the results, we can see that fib.8 and fib.9 groups in the disease group were significantly more than those in the normal group, and fib8 highly expressed TGFB1 and Fib9 highly expressed FGF7 in the disease group, which was consistent with the results of scRNA-seq.

1. Kumar, A.; D'Souza, S. S.; Moskvin, O. V.; Toh, H.; Wang, B.; Zhang, J.; Swanson, S.; Guo, L. W.; Thomson, J. A.; Slukvin, II, Specification and Diversification of Pericytes and Smooth Muscle Cells from Mesenchymoangioblasts. *Cell reports* 2017, *19* (9), 1902-1916.

2. Fu, W.; Chen, S.; Yang, R.; Li, C.; Gao, H.; Li, J.; Zhang, X., Cellular features of localized microenvironments in human meniscal degeneration: a single-cell transcriptomic study. *eLife* 2022, *11*.

3. Gan, Y.; He, J.; Zhu, J.; Xu, Z.; Wang, Z.; Yan, J.; Hu, O.; Bai, Z.; Chen, L.; Xie, Y.; Jin, M.; Huang, S.; Liu, B.; Liu, P., Spatially defined single-cell transcriptional profiling characterizes diverse chondrocyte subtypes and nucleus pulposus progenitors in human intervertebral discs. *Bone research* 2021, *9* (1), 37.

4. Litviňuková, M.; Talavera-López, C.; Maatz, H.; Reichart, D.; Worth, C. L.; Lindberg, E. L.; Kanda, M.; Polanski, K.; Heinig, M.; Lee, M.; Nadelmann, E. R.; Roberts, K.; Tuck, L.; Fasouli, E. S.; DeLaughter, D. M.; McDonough, B.; Wakimoto, H.; Gorham, J. M.; Samari, S.; Mahbubani, K. T.; Saeb-Parsy, K.; Patone, G.; Boyle, J. J.; Zhang, H.; Zhang, H.; Viveiros, A.; Oudit, G. Y.; Bayraktar, O. A.; Seidman, J. G.; Seidman, C. E.; Noseda, M.; Hubner, N.; Teichmann, S. A., Cells of the adult human heart. *Nature* 2020, *588* (7838), 466-472.

5. Smyth, L. C. D.; Rustenhoven, J.; Scotter, E. L.; Schweder, P.; Faull, R. L. M.; Park, T. I. H.; Dragunow, M., Markers for human brain pericytes and smooth muscle cells. *Journal of chemical neuroanatomy* 2018, *92*, 48-60.

6. Eilken, H. M.; Diéguez-Hurtado, R.; Schmidt, I.; Nakayama, M.; Jeong, H. W.; Arf, H.; Adams, S.; Ferrara, N.; Adams, R. H., Pericytes regulate VEGF-induced endothelial sprouting through VEGFR1. *Nature communications* 2017, *8* (1), 1574.